



# African Dust Particles over the Western Caribbean Part I: Impact on air quality over the Yucatan Peninsula

Carolina Ramírez-Romero[1], Alejandro Jaramillo[1], María F. Córdoba[1,2], Graciela B. Raga[1], Javier Miranda[3], Harry Alvarez-Ospina[4], Daniel Rosas[5], Talib Amador[5], Jong Sung Kim[6],  Jacqueline Yakobi-Hancock[6], Darrel Baumgardner[7], and Luis A. Ladino[1,*]

[1]Centro de Ciencias de la Atmósfera, Universidad Nacional Autónoma de Mexico, Mexico City, Mexico.
[2]Posgrado en Ciencias Químicas, Universidad Nacional Autónoma de Mexico, Mexico City, Mexico.
[3]Instituto de Física, Universidad Nacional Autónoma de Mexico, Mexico City, Mexico.
10    [4]Facultad de Ciencias, Universidad Nacional Autónoma de Mexico, Mexico City, Mexico.
[5]Universidad Autónoma de Yucatan, Merida, Yucatán, Mexico.
[6]Dalhousie University, Halifax, Nova Scotia, Canada.
[7]Droplet Measurement Technologies, Colorado, USA
*Correspondence to: L.A.L (luis.ladino@atmosfera.unam.mx)

**Abstract.** On a global scale, African dust is known as one of the major sources of mineral dust particles as they can be efficiently transported to different parts of the planet. Several studies have suggested that the Yucatan Peninsula could be influenced by such particles, especially in July, associated with the strengthening of the Caribbean low level jet. Although these particles have the potential to impact the local air quality significantly, as shown elsewhere (especially particulate matter, PM), the arrival and the impact of African dust into Mexican territory has not been quantitatively reported to date.

Two short-term field campaigns were conducted to confirm the arrival of African dust onto the Yucatan Peninsula in July 2017 and July 2018 at the city of Merida atmospheric observatory (20.98N 89.64W). Aerosol particles were monitored at the ground level by different on-line and off-line sensors. Several $PM_{2.5}$ and $PM_{10}$ peaks were observed during both sampling periods, with a relative increase in the PM levels ranging between 200% and 500% with respect to the normal background. Given that these peaks were found to highly correlate with super micron particles and chemical elements typically found in mineral dust particles, such as Al, Fe, Si, and K, they are linked with African dust. This conclusion is supported by combining back trajectories with vertical profiles from radiosondes, reanalysis, and satellite images to show that the origin of the air masses arriving at Merida was the Saharan Air Layer (SAL). The good agreement found between the measured $PM_{10}$ concentrations and the estimated dust mixing ratio content from MERRA-2 (Version 2 of the Modern-Era Retrospective analysis for Research and Applications) corroborates the conclusion that the degradation of the local (and likely regional) air quality in Merida is a result of the arrival of African dust.


## 1. Introduction

The second largest natural contribution of atmospheric particles, worldwide, after sea spray, is mineral dust (Pey et al., 2013). Although volcanoes and soil dust from agricultural activities are significant sources of mineral dust (Walker, 1981; Tegen et al., 2004), the largest sources are the deserts distributed around the world (Goudie



and Middleton, 2006). Africa is considered one of the most important of these sources as it emits ca. 800 Tg yr[1], corresponding to ca. 70% of the global dust (Prospero et al., 2014; Ryder et al., 2019). Therefore, African dust particles play a significant role in the climate system as they can affect the planetary radiative balance and the hydrological cycle. Their optical properties, i.e. scattering and absorption, modulate radiative forcing and as cloud condensation nuclei and/or ice nucleating particles they will impact cloud formation and evolution

(DeMott et al., 2015; Hoose and Möhler, 2012; Zhang et al., 2007). Additionally, several studies have shown that the presence of mineral dust can influence tropical cyclone formation (Dunion and Velden, 2004; Evan et al., 2006), and human health as these particles degrade air quality (Carlson and Prospero, 1972; Prospero, 1999; Venero-Fernández, 2016).

African dust particles are efficiently transported far from their emission source (Perry et al, 1997; Chiapello et al., 1997). According to Middleton and Goudie (2001), there are different trajectories that African dust experiences around the world. Among the most important, African dust particles can be transported to the western Mediterranean and Europe (Karanasiou et al., 2012; Perez et al., 2008; Prodi and Fea, 1979; Salvador et al., 2014); to the eastern Mediterranean and the Middle East (Ganor and Mamane, 1982; Ganor et al., 2010;

Athanasopoulou et al., 2016), and towards the South of the African continent (d'Almeida, 1986; Resch et al., 2008). Additionally, African dust is transported across the Atlantic Ocean to the United States, Mexico, the Caribbean region, and South America (Bravo et al., 1982; Perry et al, 1997; Chiapello et al., 1997; Prospero and Lamb, 2003; Venero-Fernández, 2016; Kramer et al., 2020). The long-range transport of African dust over the Atlantic represents 25% of the total emissions from the Saharan desert (Shao et al., 2011). This transport is

favored in the Northern hemisphere during the summer (i.e., from June to September) within a dry and hot elevated layer called the Saharan Air Layer (SAL) (Carlson and Prospero, 1972; Prospero and Carlson, 1972; Karyampudi and Carlson, 1988; Tsamalis et al., 2013; Weinzierl et al., 2016).

During the summer, the SAL ascends to altitudes between 5-7 km through interactions with cool marine air

masses (Adams et al., 2012; Chouza et al., 2016; Korte et al., 2018). Dunion and Velden (2004), Dunion and Marron (2008), and Dunion (2011) studied the characteristics of the air masses that reach the North Atlantic and the Caribbean region during the boreal summer months. They found that there are three distinct air masses: moist tropical (MT), Saharan air layer (SAL), and mid-latitude dry air intrusions (MLDAIs). Each type of air mass is associated with unique thermodynamic and kinematic characteristics, and have a wide range of possible

origins. However, the SAL and MLDAI air masses have distinct flow patterns across the North Atlantic, which allows differentiating these masses by tracking their origin. In contrast, their distinctly unique moisture characteristics allow differentiating the MT from SAL air masses (Dunion, 2011).

There are different methods for the detection of the long-range transport of African dust and its presence in

different regions around the world. For several decades the tracking of dust events has been studied by remote sensing (Chiapello et al., 1999; Dunion and Velden, 2004; Foltz and McPhaden, 2008; Prospero et al., 2005; Liu et al., 2008; Voss and Evan, 2020). Ground- and space-based tools such as the light detection and ranging



(LIDAR) and satellite sensors (e.g., the moderate resolution imaging spectroradiometer (MODIS) and the visible infrared imaging radiometer suite (VIIRS)) provide the aerosol spatial distribution with altitude in terms

of the aerosol optical depth (AOD), mass concentration, and particle size distribution (Zhang and Reid, 2006; Jackson et al., 2013). Additionally, the cloud-aerosol lidar and infrared pathfinder satellite observations (CALIPSO) can quantify (in three dimensions) the trans-Atlantic transport of African dust (Liu et al., 2008; Adams et al., 2012; Chouza et al., 2016; Korte et al., 2018).

Another useful tool is the reanalysis from global climate models that assimilates, in a statistically optimal way, satellite and ground observations. The reanalysis produces continuous, four-dimensional fields of different atmospheric variables of interest, contrasting with the observations that may be spatially and temporally sparse (Cohn, 1997; Kalnay, 2003; Rienecker et al., 2011; Schutgens et al., 2010). The use of reanalysis, considering its inherent uncertainties, has become an essential tool in the atmospheric research community (Gelaro et al.,

2017). For example, the hybrid single-particle lagrangian integrated trajectory (HYSPLIT) model has been successfully used to track the transport of African dust particles (e.g., Ashrafi et al., 2014; Prospero et al., 2005). HYSPLIT uses meteorological data from different modeling sources, including the NCEP-NCAR National Centers for Environmental Prediction (NCEP)–National Center for Atmospheric Research (NCAR) reanalysis model (Stein et al., 2015).


The transport of African dust can also be evaluated with NASA's Global Modeling and Assimilation Office (GMAO) MERRA-2 reanalysis. MERRA-2 (Version 2 of the Modern-Era Retrospective analysis for Research and Applications) is the first multidecadal reanalysis that assimilates both meteorological and aerosol data from various ground- and space-based remote sensing sources (Gelaro et al., 2017; Randles et al., 2017). Despite

some deficiencies, previous studies have demonstrated that the MERRA-2's aerosol assimilation system does indeed show considerable skill in simulating numerous observable aerosol properties (e.g., Buchard et al., 2015, 2016, 2017; Randles et al., 2017). MERRA-2 has been previously used to study the effects of aerosol particles in the earth system, in several studies focused on dust-related phenomena. For example, Buchard et al. (2017) showed the benefit of the MERRA-2 assimilation for the retrieval of the seasonality, vertical distribution, and

magnitude of the dust surface concentrations during an episode of dust transport from Africa to the Caribbean. Later on, Veselovskii et al. (2018) showed the consistency of the MERRA-2 aerosol products with MIE-Raman lidar observations performed in West Africa during a smoke and dust mixing event. Similarly, Grogan and Thorncrof (2019) studied the characteristics of African easterly waves and their relationship with synoptic-scale plumes of Saharan mineral dust. More recently, Bibi et al. (2020) studied atmospheric dust load and deposition

fluxes along the North-African coast of the Mediterranean Sea; and Aldhaif et al. (2020) studied dust events impacting the United States East Coast.

The *in situ* monitoring of aerosol properties, such as aerosol size and mass distribution, is very useful to determine their influence on local air quality and human health (Querol et al., 2019). Hence, different studies

have been carried out in the Caribbean islands and Florida to quantify the impact of African dust on the local



air quality (Prospero, 1999; Prospero and Mayol-Bracero, 2013; Prospero et al., 2014). In Barbados, the monitoring of the atmospheric aerosol mass began in 1965, while in Miami, Florida, it began in 1974 and continues to the present (Prospero and Mayol-Bracero, 2013). In Barbados, it is estimated that 50% of the $PM_{2.5}$ (i.e., with an aerodynamic diameter d<2.5 μm) and ca. 90% of the $PM_{10}$ (i.e., d<10 μm) consist of African dust

(Li-Jones and Prospero, 1998; Prospero et al., 2001; Reid et al., 2003a). In Miami, the mean daily mass concentration of mineral dust during the summer typically ranges between 10 and 100 μg m$^{-3}$, with a large interannual variability (Prospero et al., 2001). During the Puerto Rico Dust Experiment (PRIDE) campaign carried out between 28 June and 24 July, 2000, the mineral dust concentration at the ground level was found to exceed 70 μg m$^{-3}$ (Reid et al., 2003b). In the aforementioned studies, the African dust particles transported over

the Atlantic affected the local air quality, exceeding the World Health Organization (WHO) guidelines for $PM_{2.5}$ and $PM_{10}$ (i.e., above 25 μg m$^{-3}$ for $PM_{2.5}$ and 50 μg m$^{-3}$ for $PM_{10}$). According to the WHO, air pollution and its effects are considered a global health priority (WHO, 2002). Several studies have linked high concentrations of mineral dust (in terms of $PM_{2.5}$ and $PM_{10}$) to brain, cardiovascular, and respiratory diseases (Wilker et al., 2015; Brook et al., 2010; Dominici et al., 2006). According to Goudie (2014) and Zhang et al. (2016), inhaled dust

particles can cause damage to the lungs, in addition to other parts of the body such as the heart, skin, and brain. For example, in Rome, Italy, the effects of $PM_{10}$ on cerebrovascular diseases were found to increase by 5% due to the presence of African dust particles (Alessandrini et al., 2013). Similarly, Rodríguez-Cotto et al. (2013) found that $PM_{2.5}$ and $PM_{10}$ can cause asthma and allergic reactions mainly to children during the epochs of African dust in Puerto Rico. Additionally, African dust particles have been found to serve as carriers for

biological material. Griffin et al. (2001) reported the presence of viable bacteria and fungi associated with the arrival of African dust over the U.S. Virgin Islands. Similarly, Rodriguez-Gomez et al. (2020) found a higher concentration of viable bacteria and fungal propagules during summer than during winter in the Yucatan Peninsula, with summer being the season when African dust intrusions are more frequent.

The chemical and mineralogical composition of particles also plays an important role in the identification of African dust in the receptor regions (Nenes et al., 2014). The most abundant minerals present in these particles are silicates (quartz), clay minerals (kaolinite, illite, chlorite, palygorskite), feldspars (albite, anorthite) and carbonates (calcite) (Goudie and Middleton, 2006; Querol et al., 2019; Broadley et al., 2012). The major oxides in Saharan dust are $SiO_2$, $Al_2O_3$, $Fe_2O_3$, $CaO$, $MgO$ and $K_2O$ and to a lesser extent $P_2O_5$ and $TiO_2$ (Goudie and

Middleton, 2006; Linke et al., 2006). Several studies in the Caribbean have identified high levels of Fe and Al in dust events (Prospero et al., 2001; Rosinski et al., 1988). Additionally, Rosinski et al. (1988) reported high percentages of Si and Mg in particles collected in the Gulf of Mexico (GoM) during July.

Although the arrival of African dust in Mexico has been suggested for decades (e.g., Bravo et al., 1982;

Prospero, 1999; Lenes et al., 2012), to our knowledge, there has not been a comprehensive study, published in the open, peer-reviewed literature, that documents this atmospheric phenomenon. For the first time, in this study we document the arrival of African dust in the Yucatan Peninsula for two consecutive years (i.e., 2017 and





2018) using *in situ* and remote sensing measurements, reanalysis, back trajectory analysis, and complementary meteorological observations.


## 2. Materials and Methods

### 2.1 Sampling site and field campaigns

The Yucatan Peninsula is located in the southeast of Mexico. It borders with the GoM to the north, the Atlantic Ocean to the east, and the Caribbean Sea, Guatemala, and Belize to the south. The Yucatan has characteristics
that are unique to this region (Plasencia, 1998). For example, its uniform terrain, the absence of rivers, and the type of soil, formed by Cretaceous sediments that do not present mineralization and are rich in calcium, commonly called "Laja de Yucatán" (Plasencia, 1998) sets the Yucatan aside from other regions of Mexico. The average temperature of the Yucatan Peninsula ranges from 25℃ to 35℃ (World Resource Institute, 2018) with an average annual relative humidity of 79% (INEGI, 2009). The Peninsula has a warm,
semi-dry climate on the coast and a warm, sub-humid climate throughout the rest of the region, with a rainy season between summer and autumn (June-October) (Orellana et al., 2009; ProAire, 2018). Precipitation in this region is mainly due to convective activity and it is influenced by the moisture advection by the trade winds (Orellana et al., 2009; ProAire, 2018). *In situ* measurements were made in the city of Mérida, situated in the northeast sector of the Yucatan Peninsula (20.98 °N, 89.64 °W), that is the capital of the Yucatan state. Mérida
has 892,363 inhabitants (INEGI, 2015) and is 23 km away from the coast. The most representative activities in the region are tourism, commerce, and the textile industry (INEGI, 2017).

Aerosol particles were continuously monitored with sensors installed at the School of Chemistry of the Universidad Autónoma de Yucatán (FC-UADY), located in the central-western part of the city (Figure 1), as
part of the African Dust and Biomass Burning Over Yucatan (ADABBOY) project. Table 1 lists the instrumentation used to characterize particle physical, optical and chemical properties. The Partisol and MiniVol were installed on the rooftop of the FC-UADY while the other instruments were maintained in an environmentally controlled area where they sampled from inlets connected to a ventilated chimney that extended approximately 1.5 m above the roof. This measurement site is part of the University Network of
Atmospheric Observatories (RUOA) supported by the National University of Mexico (UNAM). Two intensive sampling periods were conducted between July 11 - July 31, 2017 and June 30 - July 17, 2018.

### 2.2. Aerosol concentration and particle size distribution

The particulate mass concentration was monitored continuously with $PM_{2.5}$ and $PM_{10}$ analyzers providing real-
time measurements (FH 62 C14 Thermo Scientific Inc) with a temporal resolution of one minute at a sampling flow rate of 16.7 L min$^{-1}$ (Thermo Fisher Scientific Inc, 2007).

The total number concentration of particles with sizes approximately larger than 50 nm was measured by a condensation particle counter (CPC 3010, TSI) at a sampling rate of 1 hz with a flow rate of 1.0 L min$^{-1}$ and



the aerosol number concentration as a function of particle size was monitored by an optical particle counter
        (LasAir II 310A, MSP). The LasAir has six different size bins (0.3, 0.5, 1.0, 5.0, 10.0, and 25 μm), a flow rate
        of 28.3 L min$^{-1}$ and a time resolution of 11 s.

### 2.3 Aerosol collection and chemical composition analysis

PM$_{2.5}$ and PM$_{10}$ aerosol particles were collected for 24 h with a Partisol model 2525 (Thermo Fisher Scientific
        Inc.) and for 48 h with a Minivol (3380, Air metrics) on 47 mm teflon filters (Pall Science). The MiniVol and
        Partisol flow rates were 5.0 L min$^{-1}$ and 16.7 L min$^{-1}$, respectively. After the sampling periods, the filters were
        placed in 60 mm Petri dishes and stored at 4°C prior to the chemical analysis.

Elemental analysis was performed on each filter using X-Ray Fluorescence (XRF) with the X-ray spectrometer
        at Laboratorio de Aerosoles, Instituto de Fisica, UNAM (Espinosa et al., 2012). The X-ray tube was made by
        Oxford Instruments (Scotts Valley, CA, USA), an Rh anode and an Amptek X-123SDD spectrometer (Bedford,
        MA, USA) were used. The samples were irradiated for 900 s working with a current of 500 μA and resulting in
        a spectrum that was analyzed using the WinQXAS computer code (IAEA, 1997). The product of this analysis
derived mass concentrations of Fe, Al, Si, Ca, Na, P, Mg, Mn, Ti, Cl, P, Zn, K, S, Cu, and Ni along with their
        associated uncertainties, as described by Espinosa et al. (2010).

### 2.4 Meteorological and satellite data

        The local and regional meteorological conditions were monitored using different approaches. The RUOA
meteorological sensors were placed at the rooftop of the FC-UADY (Table 1) continuously measuring the wind
        speed and direction, air temperature, relative humidity, solar radiation, and precipitation. To derive the regional
        and vertical distribution of meteorological conditions, radiosondes and reanalysis were used. The information
        provided by the radiosondes launched was from three World Meteorological Organization (WMO) stations in
        the Yucatan Peninsula, as shown in Figure 1. These locations are: Merida (Mérida International Airport,
Mexico. (WMO index: 76644), Cancún (WMO index: 76595), and Belize (Philip SW Goldson International
        Airport, Belize. WMO index: 78583). The processed radiosonde data is obtained from the University of
        Wyoming (http://weather.uwyo.edu/upperair/sounding.html).

        Hourly total precipitable water vapor and three-dimensional 3-hourly aerosol mixing ratio data were obtained
from the MERRA-2 reanalysis (GMAO, 2015a, 2015b). The aerosol properties in MERRA-2 were simulated
        with the Goddard Chemistry Aerosol Radiation and Transport model (GOCART), which takes into account the
        sources, sinks, and chemistry of 15 externally-mixed aerosol mass mixing ratio tracers: dust (five non-
        interacting size bins), sea salt (five non-interacting size bins), hydrophobic and hydrophilic black and organic
        carbon (BC and OC, respectively; four tracers), and sulfate (SO$_4$) (Randles et al., 2017; Buchard et al., 2017).


        The air mass back trajectories were calculated using the HYSPLIT model from the National Oceanic and
        Atmospheric Administration (NOAA). In conjunction with the *in situ* measurements, the back trajectories were



calculated considering the maximum concentration of PM reported by the $PM_{2.5}$ and $PM_{10}$ analyzers. The trajectories were initiated at 50, 250 and 500 m above ground level going backward in time for 13 days.

Although Kramer et al. (2020) reported that mineral dust particles arrive in Miami ca. 10 days after they are produced in North Africa, they may take a longer time to reach the Yucatan Peninsula. The different heights of the HYSPLIT back-trajectories runs were chosen to determine the rate of descent of dust air masses to the surface. Also considered was the aerosol optical depth (AOD) measured with MODIS instruments in the Aqua and Terra satellites.

## 3. Results and Discussion

### 3.1. Local evidence

Several studies have shown that air quality ($PM_{2.5}$ and $PM_{10}$) significantly deteriorates upon the arrival of African dust plumes (Prospero and Lamb, 2003; Prospero et al., 2001; Prospero and Mayol-Bracero, 2013; Prospero et al., 2014). Figure 2 shows the time series of the $PM_{2.5}$ and $PM_{10}$ concentrations for the July-August period of 2017 and 2018. Some high concentration PM peaks are clearly identified, with $PM_{2.5}$ and $PM_{10}$ values as high as 54 $\mu g$ $m^{-3}$ and 135 $\mu g$ $m^{-3}$, respectively. Henceforth those peaks will be referred to in our study as African dust peaks (ADPs). Note that the background concentration of $PM_{2.5}$ is ~ 4 $\mu g$ $m^{-3}$ and of $PM_{10}$ is ~ 10 $\mu g$ $m^{-3}$. The ADPs found in 2017 (i.e., July 22-24, 27-28, and August 4, 6-7) resulted in an increase of 300% in $PM_{2.5}$ and 500% in $PM_{10}$ with respect to the background. In 2018, the ADPs (i.e., July 10-11, 13-15, 16-17, 23 -26, and August 9-10) exceeded 200% and 300% of the background levels of $PM_{2.5}$ and $PM_{10}$, respectively. The aforementioned ADPs not only exceeded the $PM_{2.5}$ and $PM_{10}$ thresholds suggested by the WHO (i.e., $PM_{2.5}$=25 $\mu g$ $m^{-3}$ and $PM_{10}$=50 $\mu g$ $m^{-3}$, 24-h mean) but more than double them, as was the case for the August 9-12, 2018 event. Similar behavior has been previously observed in Puerto Rico, Miami, and Barbados during the arrival of African dust particles (Reid et al., 2003b; Prospero et al., 2005, 2014). The mass concentrations of $PM_{2.5}$ and $PM_{10}$ were found to be 49% and 54% higher in 2018 than in 2017, respectively, suggesting a higher frequency or intensity of African dust plumes arriving over Merida in 2018.

Figure 2 also shows the elemental composition obtained from the XRF analysis (16 elements) for five ADPs observed during the 2017 and 2018 field campaigns. In addition, one day from each field campaign was selected to determine the elemental background composition. The selected days are July 10-11 in 2017 and July 6 in 2018. These days were chosen because the $PM_{2.5}$ and $PM_{10}$ concentrations were within the mean background values and did not include any atypical peak nor apparent external influence.

High levels of sodium (Na, pink), chlorine (Cl, turquoise blue), sulfur (S, dark orange), and calcium (Ca, light green) were found in the background samples, corresponding to >70% of the total mass. The presence of Na and Cl are expected in airborne particles at this site given the city's proximity to the GoM (i.e., 23 km away). Cerón et al. (2002) reported large concentrations of Na, Cl, and Mg that originated from sea salt, when analyzing the composition of rainwater from the Yucatan Peninsula. The high levels of S can be associated with local


anthropogenic activities such as vehicular, ship, and industrial emissions (e.g., Corbett and Fischbeck, 1997; Cerón-Bretón et al., 2018). Additionally, given the short distance between Merida and the GoM, it is possible that dimethylsulfide (DMS) production from plankton in the GoM can be a natural source of S, as has been shown in other studies (e.g., Rosinski et al., 1988; Kloster et al., 2006; Vallina and Simó, 2007). Finally, the presence of Ca could be related to the limestone soil prevalent in the Yucatan Peninsula and the resuspension

of road dust (Plasencia, 1998; Querol et al., 2019).

Interestingly, the elemental composition of the airborne particles collected during the ADPs showed higher concentrations of silica (Si, dark yellow), aluminum (Al, light purple), and iron (Fe, dark purple) than those in the background particles. While the Si concentrations are approximately three times larger than the baseline, Al

and Fe increased by eight and 12 times, respectively. To corroborate the relationship between the increase in PM and the African dust, each of the 16 elements analyzed by XRF were correlated with the $PM_{2.5}$ and $PM_{10}$ concentrations. Out of the 16 elements Al, Si, K, and Fe were the only ones with correlation coefficients r>0.6 ($p < 0.05$) for both years, as shown in Figure S1. The present results are in agreement with previous studies that showed high correlation coefficients between the aforementioned elements (e.g., Caquineau et al., 1998; Guieu

et al., 2002; Trapp et al., 2010). Also, the concentration of aerosol particles with diameters between 0.5 μm and 25 μm, as measured by the LasAir, were found to highly correlate with the $PM_{2.5}$ and $PM_{10}$ concentrations, r=0.79 and r=0.87, respectively (Figure S2). Finally, the typical background particle size distribution showed significant changes during the arrival of ADPs for particles ranging between 0.5 μm and 5.0 μm (Figure S3). It is widely known that the typical size of African dust particles transported over long distances ranges from 0.1

μm to 20 μm (e.g., Bégue et al., 2012; Denjean et al. 2016). Overall, the high concentration of coarse particles and the increase of Al, Si, and Fe during the ADPs, together with the good correlations found between the $PM_{2.5}$ and $PM_{10}$ concentrations with Al, Si, K, Fe, and particles larger than 0.5 μm strongly suggests that the ADPs are mineral particles associated with dust transported from Africa to Mexico.

Additionally, it is important to note that Al, Si, K, and Fe are common oxides found in African dust composed of minerals and clays such as: Quartz ($SiO_2$), Kaolinite ($Al_2Si_2O_5(OH)_4$), Illite $(K,H_3O)(Al,Mg,Fe)_2(Si,Al)_4O_{10}[(OH)_2,(H_2O)]$, Chlorite $((MgFe)_5Al)(AlSi_3)O_{10}(OH)_8$, Palygorskite $(Mg,Al)_2Si_4O_{10}(OH)\cdot 4(H_2O)$, and Feldspars such as albite ($NaAlSi_3O_8$), Anorthite ($CaAl_2Si_2O_8$) and orthoclase $KAlSi_3O_8$, among others (A. Goudie and Middleton, 2006; Linke et al., 2006; Broadley et al., 2012; Querol et

al., 2019). Rosinski et al. (1988) reports that up to 90% of the collected airborne particles in the presence of dust events in the GoM contained Al, Fe, Si. In Puerto Rico, Reid et al., (2003b) found that during dust events that reached the island, the concentrations of Si and Al on aerosol particles (> 0.74 μm) were above 10 μg m$^{-3}$ and 5 μg m$^{-3}$, respectively. Similarly, Prospero et al., (2001) reports concentrations of Al > 1.0 μg m$^{-3}$ and Fe > 0.5 μg m$^{-3}$ on $PM_{2.5}$ on days when high concentrations of African dust particles were reported in Miami.






To confirm that no aerosol sources other than the African dust were the origin of the high PM peaks observed in Merida in July 2017 and 2018, the $PM_{2.5}$ and $PM_{10}$ concentrations were correlated with other measured variables. As shown in Figures S2 and S4, $PM_{2.5}$ and $PM_{10}$ concentrations are poorly correlated ($r<0.09$) with local pollution emissions such as pPAHs, black carbon (inferred from the absorption coefficient), and NOx.

Note that those gases and particles can be considered as proxies of anthropogenic pollutants generated by the incomplete combustion of fossil fuels and biomass burning, as previously demonstrated for Merida (Muñoz-Salazar et al., 2020; Alvarez-Ospina et al., 2020). Also, correlation coefficients below 0.29 were found between $O_3$ and solar radiation with the $PM_{2.5}$ and $PM_{10}$ concentrations. Muñoz-Salazar et al., (2020) found in Merida that ultrafine aerosol particles of secondary origin are correlated with $O_3$, a proxy of photochemical activity and

hence, of secondary particle production. Therefore, it is very unlikely that secondary organic particles could be the source of the ADPs observed in Merida.

Finally, although none of the different meteorological variables monitored at the surface level were found to correlate with the $PM_{2.5}$ and $PM_{10}$ concentrations, as shown by the wind roses in Figure S5, easterly winds were

prevalent when ADPs were observed. This is relevant since African dust can only be transported by easterly winds.

### 3.2. Larger scale observations

To evaluate the source of the ADPs observed in Merida from a large-scale perspective, we focus on the

classification of tropical air masses in the North Atlantic and the Caribbean region during the boreal summer months proposed by Dunion (2011). We used HYSPLIT to estimate the trajectories of different air masses that reached Merida in the periods of July-August 2017 and 2018. HYSPLIT trajectories for the 2017 and 2018 ADPs point to an African origin and, therefore, suggest that these air masses are either MT or SAL (See Figure S6). To differentiate the MT from SAL, we focused on their distinctly unique moisture characteristics. Dunion

(2011) proposes that a threshold of 45 mm of total precipitable water vapor (PWV), which corresponds to the total amount of water vapor contained in the atmospheric column from the surface to the top of the troposphere (AMS, 2000), can be used to differentiate dry from moist air masses. This value is consistent with other studies that use PWV to identify dry-air days (e.g., Hankes and Marinaro, 2016), and since deep tropical convection begins to increase above a critical PWV value of 50 mm (Holloway and Neelin, 2009).


Figure 3 shows the time series of PWV for the July-August 2017 and 2018 periods at each WMO radiosonde site. The black solid line shows PWV from MERRA-2 (GMAO, 2015a), together with PWV estimated from the available radiosonde profiles shown as the dashed blue lines. One caveat is that in the periods of interest, there is a striking lack of radiosonde data. Nevertheless, we can see a good agreement between the available

observed PWV and that of PWV from MERRA-2. Therefore, the latter can be used as a good approximation for PWV in the region to differentiate moist from dry air masses. In Figure 3, the periods where PWV is less than 45 mm are highlighted in red. These periods show dry air masses that coincide with air mass trajectories with an African origin (i.e., in 2017: July 22-24, July 27-28, August 04, and August 6-7; and in 2018: July 10-





12, July 13-15, July 16-17, July 23-26 and August 9-12), allowing us to conclude that these dry air masses have
mainly SAL characteristics.

The arrival of African dust in Merida was also explored from the MERRA-2 dataset. Figure 4 shows the time
series of the estimated vertical profiles of the 3-hour time series of dust mixing ratio from MERRA-2 at Merida
for 2017 and 2018. It shows that the events corresponding to the arrival of dry air masses from Africa shown in
Fig. 3 nicely correlate with high dust mixing ratios, strongly supporting the hypothesis of SAL air reaching the
Yucatan Peninsula. Figure 4 also shows that the July-August period of 2018 was particularly active with
frequent arrivals of dust in the region, in agreement with the higher $PM_{2.5}$ and $PM_{10}$ concentrations measured in
2018, as depicted in Fig. 2. Figure S7 focuses on the vertical profiles of dust mixing ratio for the periods of July
21 to 25, 2017 and July 12 to 16, 2018. These periods show the increase of dust in the atmospheric column in
the studied region, supporting the hypothesis that the source of the ADPs shown in Figure 2 is likely African
dust.

Finally, the arrival of African dust plumes over the Yucatan Peninsula was confirmed by investigating the AOD
detected by the MODIS Aqua/Terra satellite for July 2017 and 2018, as shown in Figures S8 and S9. Although
this information cannot be used to perform quantitative analysis, the AOD images allow us to confirm the arrival
of African dust plumes into the Yucatan Peninsula. Additionally, as in Figures 2, 3, and 4, the AOD images
also show that the African dust plumes activity was higher in 2018 than in 2017. Previous studies have also
used the AOD from MODIS to identify the arrival of African dust. For example, Koren et al. (2006) tracked
the long-range transport of dust from the Bodélé depression (North-Central Africa) to the Amazon basin.
Similarly, Kalashnikova and Kahn (2008) demonstrated that with the MODIS it is possible to observe the
evolution of African dust plumes over the Atlantic Ocean. Additionally, Kaufman et al., (2005) identified and
quantified the transport and deposition of mineral dust over the Atlantic Ocean using MODIS data.

### 3.3. Comparison of *in situ* observations and reanalysis

The daily mean $PM_{10}$ from MERRA-2 was estimated using the method proposed by Provençal et al. (2017).
The black line in Fig. 5 shows the estimated $PM_{10}$, which is compared to the $PM_{10}$ measured by the RUOA
station in Merida (blue line). The estimated surface dust mixing ratio from MERRA-2 is also shown in red.
Figure 5 shows that MERRA-2 overestimates compared to the ground-based measurements of $PM_{10}$.
Nevertheless, it should be clarified that the reanalysis information corresponds to a region 0.5° x 0.625°,
implying that the MERRA-2 is estimating the regional average, while the station corresponds to a local
measurement.

Despite these differences, Figure 5 shows that the observations at the RUOA station have variations similar to
those of MERRA-2. Figure 6 shows the dispersion diagram of the daily mean surface dust mixing ratio from
MERRA-2 vs. $PM_{10}$ measured from RUOA station for the periods indicated in Figure 5. It shows a high
correlation between the estimated dust and the measured $PM_{10}$ in particular for the 2018 period, which was





particularly active with constant arrivals of African dust to the region, as shown in Figure 4. A similar analysis was performed for the 3-h estimated and measured $PM_{10}$, as shown in Figures S10 and S11, with identical conclusions as for 24-h averages.


### 4. Conclusions

For the first time, the arrival of African dust into Mexican territory is quantitatively verified. The arrival of African dust particles in Merida significantly degraded the local air quality as $PM_{2.5}$ and $PM_{10}$ concentrations increased up to 500% with respect to the background. Therefore, the presence of African dust in Merida and

the Yucatan Peninsula has the potential to trigger or exacerbate several diseases, as has been reported elsewhere (Wilker et al., 2015; Brook et al., 2010; Dominici et al., 2006). The arrival of African dust in other regions of the world has led to even higher PM concentrations than those found in the present study. For example, Querol et al. (2009) reported in Spain peak concentrations of $PM_{10}$ during dust events that reached up to 250 $\mu g\ m^{-3}$. Similarly, in Nicosia (Cyprus), concentrations of $PM_{10}$ up to 470 $\mu g\ m^{-3}$ were measured (Achilleos et al., 2014).

Moreover, African dust particles can also be a serious health threat as they serve as the carrier of biological material originating in Africa. If the foreign biological particles are opportunistic pathogens, they can cause a variety of diseases in the receptor regions, such as the Yucatan Peninsula. Finally, those particles can impact the development of precipitation affecting the regional hydrological cycle when they serve as efficient ice nucleating particles (Hoose and Möhler, 2012; Murray et al., 2012; Kanji et al., 2017).


As shown in the present study, combining ground-based off-line and on-line sensors provides robust evidence of the arrival of African dust; however, we also show that the combination of back trajectories with radiosondes, and the estimated surface dust mixing ratio from MERRA-2 are powerful tools that can be exploited when *in situ* information is missing, especially in developing countries where the necessary instrumentation is scarce.


Continuous monitoring of the arrival of African dust is of high importance not only in the Caribbean islands but also at other sites in Latin America such as Mexico, Belize, Guatemala, Honduras, etc. Additionally, epidemiological and statistical studies to track down the number of hospital admissions caused by respiratory issues before and after the arrival of African dust is urgently needed in the Yucatan Peninsula. This will allow

policy-makers and local authorities to understand how strong the African dust impact is on local health and the need for better forecasting of such events.

*Data availability*. Data are available upon request to the corresponding author.

*Author contributions*. CRR, GBR, and LAL designed the field campaigns and the experiments. CRR, MFC, HAO, DR, TA, and LAL carried out the aerosol measurements. CRR and AJ analyzed the remote sensing data. JM and HAO performed the chemical analyses. GBR, DB, DR, JSK, JYH, and LAL installed the equipment



and provided the infrastructure for the ADABBOY project. CRR, AJ, and LAL wrote the paper, with contributions from all coauthors.


*Competing interest.* The authors declare that they have no conflict of interest.

*Acknowledgments.* This study was financially supported by the Consejo Nacional de Ciencia y Tecnologia (Conacyt) and the Universidad Autónoma de Yucatán through the FC-2164 and SISPROY-FQUI-2018-0003

grants, respectively. The authors thank the University Network of Atmospheric Observatories (RUOA) for providing meteorological and criteria pollution data. A. Jaramillo acknowledges the fellowship from DGAPA at UNAM. Also, the authors express their gratitude to Elizabeth Garcia, Juan Carlos Pineda, Aline Cruz, and Javier Juarez for their invaluable help and support.

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

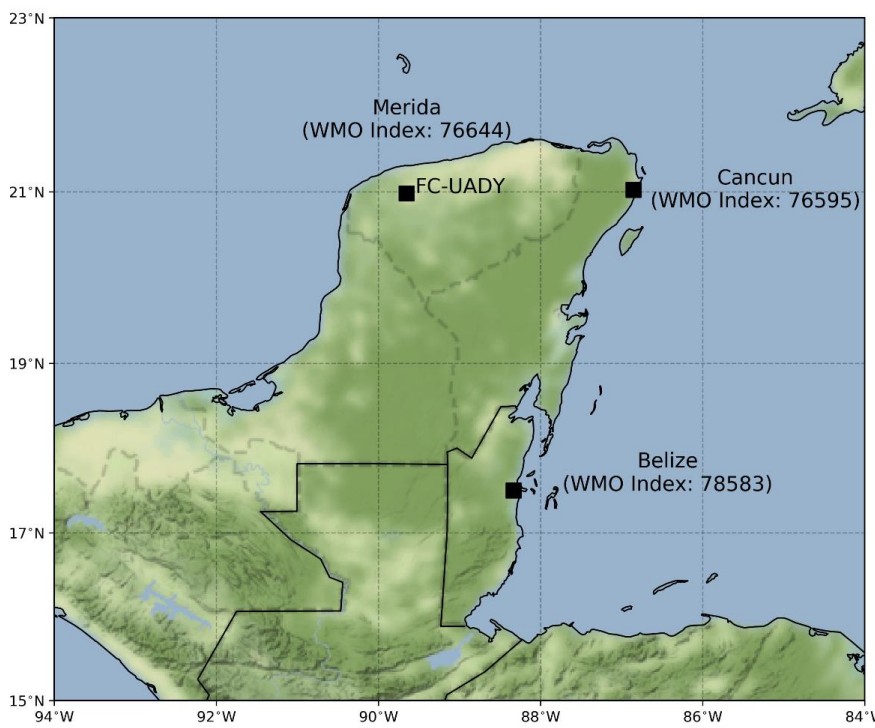

**Figure 1.** Location of the sampling site at the School of Chemistry of the Autonomous University of Yucatan (FC-UADY) and the three World Meteorological Organization (WMO) radiosonde stations located in the Yucatan Peninsula: Mérida International Airport, Mexico (WMO index: 76644); Cancun, Mexico (WMO index: 76595) and Philip S. W. Goldson International Airport, Belize (WMO index: 78583).

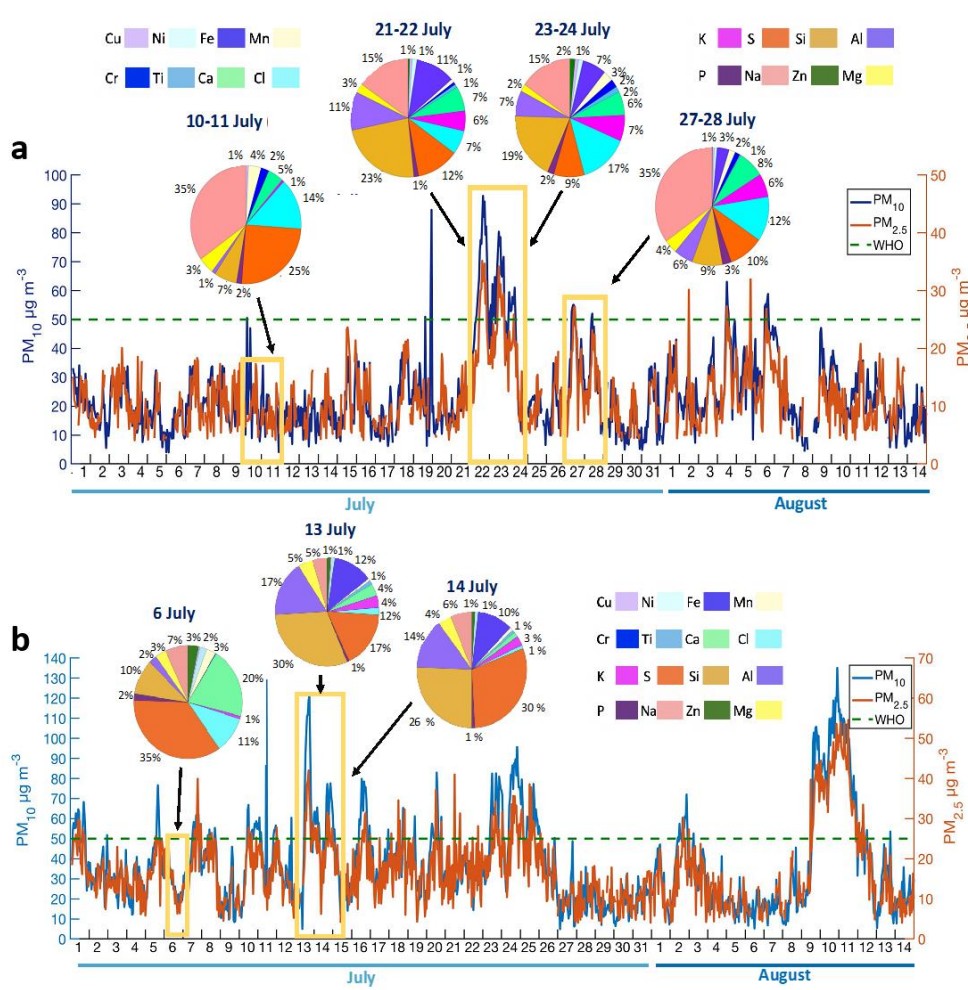


**Figure 2.** a) Mass concentrations of PM$_{2.5}$ and PM$_{10}$ in 2017, and b) in 2018. The green horizontal line depicts the World Health Organization Air Quality Guideline for 24 h mean PM$_{2.5}$ (Y axis right) and PM$_{10}$ (Y axis left) concentrations of 25 μg m$^{-3}$ and 50 μg m$^{-3}$, respectively (WHO, 2002). The pie charts represent the elemental composition from the XRF for the ADPs.




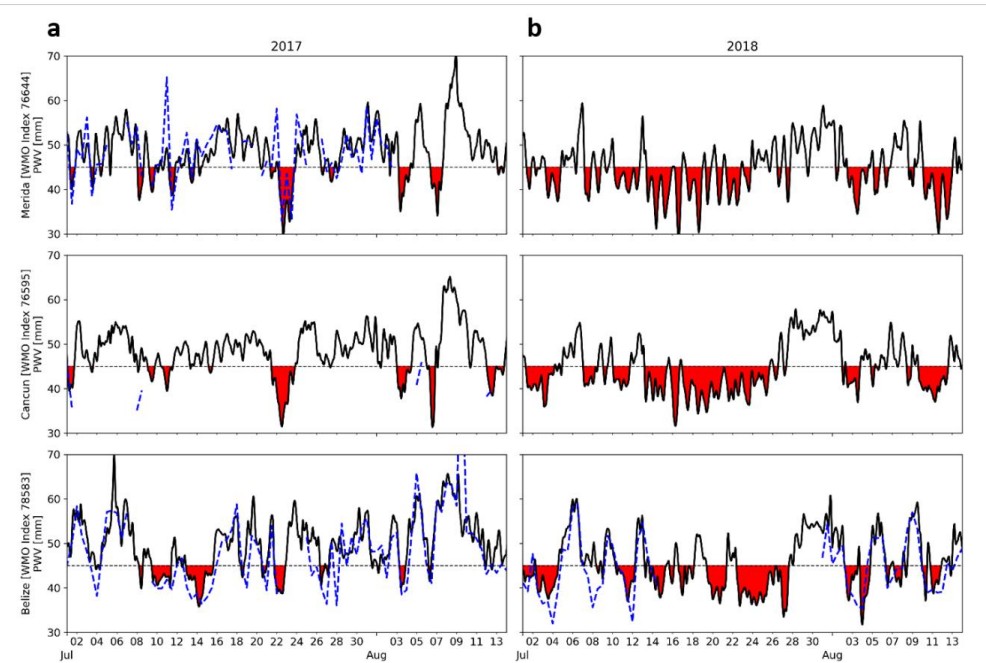

**Figure 3.** MERRA-2 precipitable water (black solid line) and estimated from radiosonde measurements (blue dashed line), for the three WMO radiosonde stations located in the Yucatan Peninsula. a) July-August 2017, and b) July-August 2018. The red areas represent the periods where PWV is less than 45 mm.

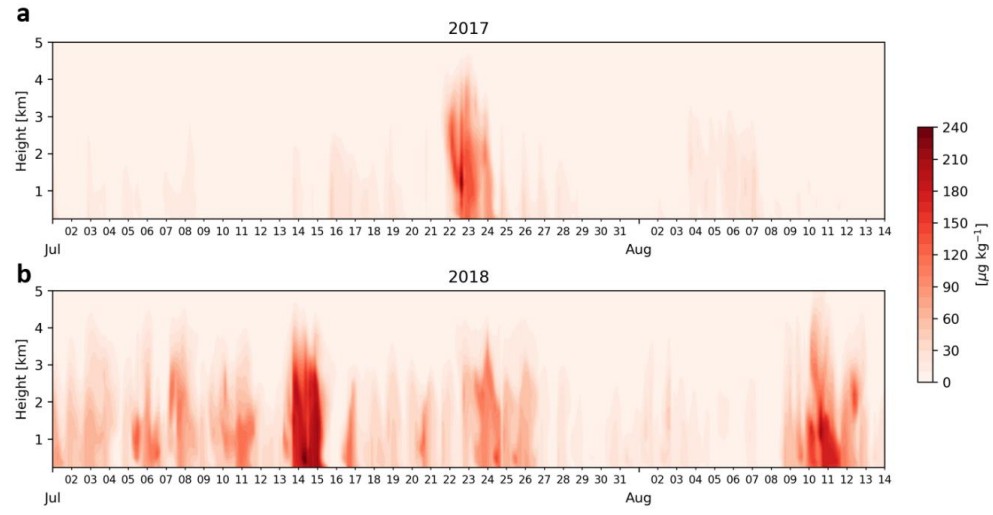


**Figure 4.** 3-hour time series of the vertical profile of the estimated dust content from MERRA-2 for the July 1 to August 14 period for a) 2017 and b) 2018 for the Merida region.


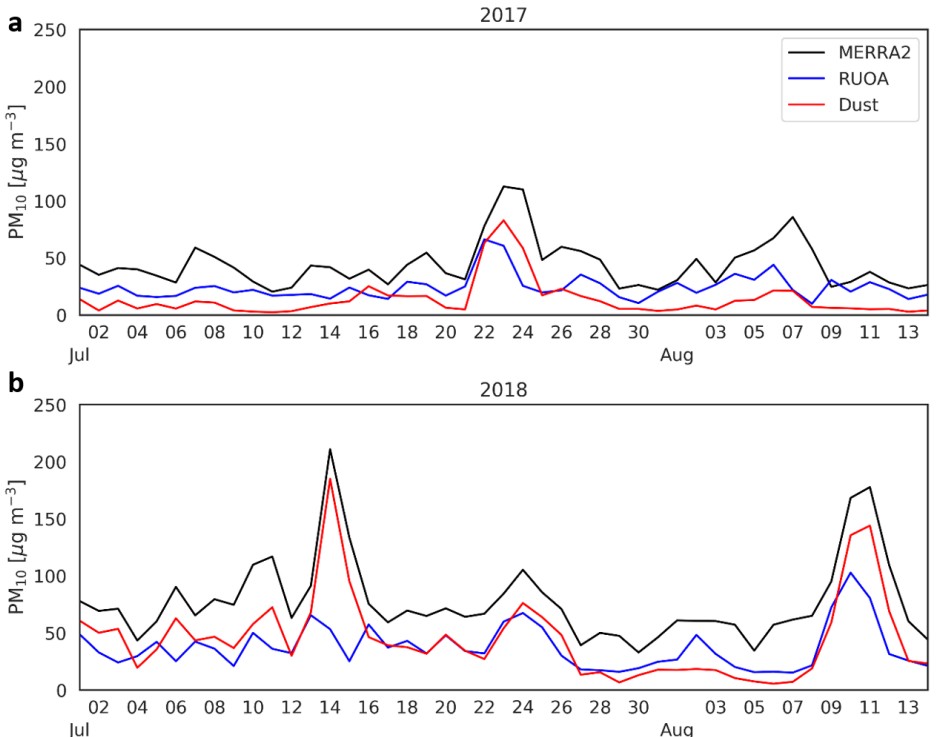

**Figure 5.** Daily mean of $PM_{10}$ estimated from MERRA-2 (black line), measured by the RUOA station (blue line), and the estimated dust mixing ratio content from MERRA-2 (red line) for a) 2017 and b) 2018.

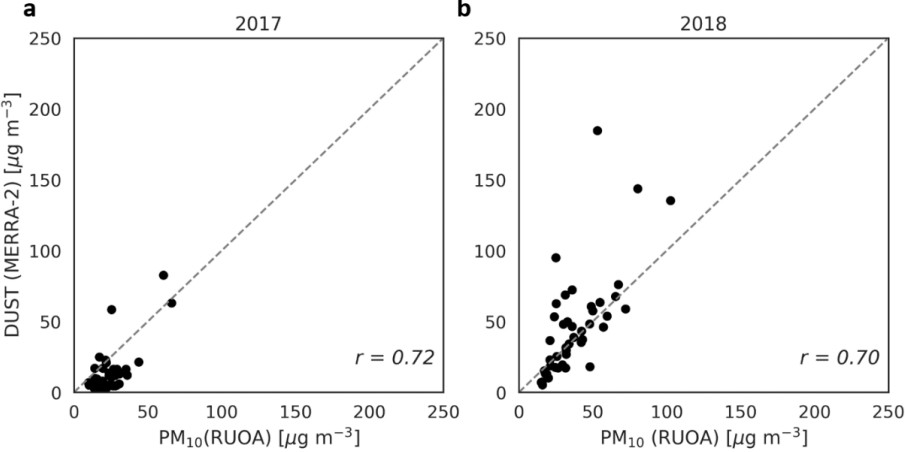

**Figure 6.** Dispersion diagrams of surface dust mixing ratio from MERRA-2 (y-axis) vs. the $PM_{10}$ from the RUOA station for the periods shown in Figure 5 for a) 2017 and b) 2018.



**Table 1**. Summary of the used instrumentation and the measured variables.

| Measured variable | Instrumentation |
|---|---|
| Particle mass concentration | $PM_{2.5}$ and $PM_{10}$ analyzer (FH 62 C14 Thermo Scientific Inc.) |
| Total particle concentration (d>50 nm) | Condensation Particle Counter (CPC, 3010, TSI) |
| Particle size distribution (d>300 nm) | Optical Particle Counter (LasAir, II 310A, MSP) |
| Aerosol collection ($PM_{10}$ and $PM_{2.5}$) | Partisol (2525, Thermo Fisher Scientific Inc) and MiniVol (3380, Air metrics) |
| Nitrogen oxides ($NO_x$) <br> Ozone ($O_3$) | $NO_x$ analyzer (42 i Thermo Scientific Inc.) <br> $O_3$ analyzer (49 i Thermo Scientific Inc.) |
| Temperature, relative humidity precipitation <br> Wind direction and wind speed <br><br> Solar radiation | T and HR Sensor ( VAISALA HMP 115) <br> Pluviometer (Texas Electronics, TR-525M) <br> Wind direction and wind speed sensor (Gill, 1405-PK-100) <br> Radiation pyrometer (Intertek, 20W) |
| Absorption coefficient <br><br> Particle-bound polycyclic aromatic hydrocarbons (pPAHs) concentration | Soot Absorption Photometer PSAP (Radiance Research) <br> Photoacoustic spectroscopy (PAS 2000, Ecochem) |