# Peer review of "African Dust Particles over the Western Caribbean Part I: Impact on air quality over the Yucatan Peninsula"

_Atmospheric Chemistry and Physics, 2020_

## Referee Comment (RC1) · Anonymous Referee #1 · 22 Jul 2020

General comment This manuscript reports on the identification of African dust transport and their role in impacting air quality in the Caribbean region (Yucatan Peninsula, Mexico). Authors have used multiple observations, including ground level PM10, 2.5 measurement, elemental characteristics of aerosols, radiosonde profiles as well as satellite observation (MODIS) and reanalyses data (HYSPLIT and MEERA-2), to decouple dusty period and normal days. Using these data, they are successful in inferring high mass loading is resulted from the African dust transport rather from local emissions. Authors claim, this is first data on aeolian dust concentration and characteristics reported from this study region. Such studies at regional scale are important towards our understanding of dust transport and their cycling. The paper is well written and I

recommend publication after addressing concerns stated in my specific comments.

Specific comments: 1. Introduction is too long and exhaustive. Several information is not needed for examples: Line 81-83: CALIPSO discussion is not needed here. There are several (Line 120-140), which can be removed and make it focussed. 2. Line 243-245: How background concentrations are estimated? 3. Line: Line 307-311: these nedd to be rephrased. Correlation between Ozone and PM infers what? Not clear. 4. Line325: How PWV is estimated both using radiosonde and MEERA-2? A detailed methodology need to be discussed in section 2. 5. Line 375-377: Figure shows r =0.72 for 2017 and r =0.70 for 2018, contrary to stated here. Please check. 6. Conclusion need to be modified significantly. Statements related to health impacts are highly speculative and cannot be stated in the conclusion. Line 384-394 can be removed or can be stated in introduction. I suggest Authors to highlight outcome of their study.
* * *

---

## Referee Comment (RC2) · Cassandra Gaston (Referee) · 4 Nov 2020

General:

This paper quantifies the intrusion of African dust in the Yucatan peninsula for the first time and quantifies the impact of this dust intrusion on air quality. This paper is very well-written and should be considered for publication after consideration of my minor comments.

Specific Comments

Introduction:

[Figure]

1. Line 45-49: also cite [J. M.; Prospero et al., 2014]

2. Line 55-56: also cite [Barkley et al., 2019; J. M.; Prospero et al., 1981]

3. Line 76: please replace Prospero et al 2005 with [J. M. Prospero et al., 2002]

4. Paragraph starting on Line 96, a recent study by Prospero et al 2020 also shows the value of MERRA-2 for tracking African dust transport.

Methods

1. Line 158: Define GoM

2. Line 229: HYSPLIT trajectories below 200 m are usually not advised due to the effects of terrain. The authors should provide justification for the 50 m HYSPLIT analysis.

Results and Discussion

1. Line 304: define pPAHs and NOx.

REFERENCES CITED

Barkley, A., et al. (2019), African biomass burning is a substantial source of phosphorus deposition to the Amazon, Tropical Atlantic Ocean, and Southern Ocean, PNAS, DOI:10.1073/pnas.1906091116.

Prospero, J. M., F.-X. Collard, J. Molinie, and A. Jeannot (2014), Characterizing the annual cycle of African dust transport to the Caribbean Basin and South America and its impact on the environment and air quality, Glob. Biogeochem. Cy., 29, 757-773.

Prospero, J. M., P. Ginoux, O. Torres, S. E. Nicholson, and T. E. Gill (2002), Environmental characterization of global sources of atmospheric soil dust identified with the Nimbus 7 Total Ozone Mapping Spectrometer (TOMS) absorbing aerosol product, Reviews of Geophysics, 40(1).

Prospero, J. M., R. A. Glaccum, and R. T. Nees (1981), Atmospheric transport of soil dust from Africa to South America, Nature, 289, 570-572.

---

## Author Comment (AC1) · 9 Nov 2020

We would like to thank the Reviewers for their positive evaluation of our manuscript and for their constructive suggestions, which helped us to improve the manuscript. Specific answers and manuscript modifications related to the Reviewer's comments are given below in red text.

**Reviewer #1:**

1. Introduction is too long and exhaustive. Several information is not needed for examples:
Line 81-83: CALIPSO discussion is not needed here.

A/ The text was shortened as follows. Lines 78-82: "Ground- and space-based tools such as the light detection and ranging (LIDAR) and satellite sensors (e.g., the moderate resolution imaging spectroradiometer (MODIS) and the visible infrared imaging radiometer suite (VIIRS)) provide the aerosol spatial distribution with altitude in terms of the aerosol optical depth (AOD), mass concentration, and particle size distribution (Zhang and Reid, 2006; Jackson et al., 2013). "

There are several (Line 120-140), which can be removed and make it focussed.

A/ We agree with the reviewer. The text was shortened and its readability was improved. The length of this paragraph was reduced from 19 lines to 14 lines in the revised manuscript as follows.

Lines 121-140:  "In Miami, the mean daily mass concentration of mineral dust during the summer typically ranges between 10 and 100 µg m$^{-3}$, with a large interannual variability (Prospero et al., 2001). During the Puerto Rico Dust Experiment (PRIDE) campaign, the mineral dust concentration at the ground level was found to exceed 70 µg m$^{-3}$ (Reid et al., 2003b). In the aforementioned studies, the African dust particles transported over the Atlantic affected the local air quality, exceeding the World Health Organization (WHO) guidelines for PM$_{2.5}$ and PM$_{10}$. According to the WHO, air pollution and its effects are considered a global health priority (WHO, 2002). Several studies have linked high concentrations of mineral dust (in terms of PM$_{2.5}$ and PM$_{10}$) to brain, skin, lung, cardiovascular, cerebrovascular, and respiratory diseases (Alessandrini et al., 2013; Goudie, 2014; Wilker et al., 2015; Brook et al., 2010; Dominici et al., 2006; Zhang et al., 2016). Additionally, African dust particles have been found to serve as carriers for biological material. Griffin et al. (2001) reported the presence of viable bacteria and fungi associated with the arrival of African dust over the U.S. Virgin Islands. Similarly, Rodriguez-Gomez et al. (2020) found a higher concentration of viable bacteria and fungal propagules during summer than during winter in the Yucatan Peninsula, with summer being the season when African dust intrusions are more frequent."

2. Line 243-245: How background concentrations are estimated?

A/ The text was modified to clarify this. Lines 244-247: "Henceforth those peaks will be referred to in our study as African dust peaks (ADPs). Note that the background concentration (defined as the lowest values within the sampling period where the chemical composition was available) of PM$_{2.5}$ is ~ 4 µg m$^{-3}$ and of PM$_{10}$ is ~ 10 µg m$^{-3}$."

3. Line: Line 307-311: these need to be rephrased. Correlation between Ozone and PM infers what? Not clear.

A/ The sentence was rephrased as follows. Lines 311-314: "Also, correlation coefficients below 0.29 were found between $O_3$ and solar radiation with the $PM_{2.5}$ and $PM_{10}$ concentrations indicating that it is very unlikely that secondary organic particles could be the source of the ADPs observed in Merida, as was the case in high particle concentration events shown by Muñoz-Salazar et al. (2020)".

4. Line325: How PWV is estimated both using radiosonde and MEERA-2? A detailed methodology needs to be discussed in section 2.

A/ We appreciate the reviewer's comment. We modified the text to clarify that the PWV is readily available from the MERRA-2 vertically Integrated Diagnostics. The PWV from the radiosonde profiles is calculated using the standard equation from AMS (2000), which is added to the revised manuscript for a better reference.

Lines 335-340: "Note that PWV is given by the vertical integral of the mixing ratio x(p) at the pressure level $p$ in the layer bounded by pressures $p1$ and $p2$ and can be calculated using Equation 1:

$$PWV = \frac{1}{\rho g} \int_{p1}^{p2} x \, dp \qquad \text{Equation (1),}$$

where $\rho$ represents the density of water and $g$ is the acceleration of gravity."

Lines 342-345: "Figure 3 shows the time series of PWV for the July-August 2017 and 2018 periods at each WMO radiosonde site. The black solid line shows PWV from MERRA-2, available from the Vertically Integrated Diagnostics (GMAO, 2015a), together with PWV estimated using Equation 1 from the available radiosonde profiles shown as the dashed blue lines."

5. Line 375-377: Figure shows r =0.72 for 2017 and r =0.70 for 2018, contrary to stated here. Please check.
A/Line 386: "high" was changed to "good" to properly reflect the meaning of the obtained correlation coefficients.

6. Conclusion needs to be modified significantly. Statements related to health impacts are highly speculative and cannot be stated in the conclusion. Line 384-394 can be removed or can be stated in introduction.

A/ The conclusions were modified, especially the lines highlighted by the reviewer. The Conclusions in the revised manuscript are as follows.

Lines 393-421: "For the first time, the arrival of African dust into Mexican territory is quantitatively verified. The arrival of African dust particles in Merida significantly degraded the local air quality as $PM_{2.5}$ and $PM_{10}$ concentrations increased up to 500% with respect to the background. Therefore, the presence of African dust in Merida and the Yucatan Peninsula could be a potential health threat to their inhabitants. Although the African dust intrusions caused an increase in particulate matter in Merida (Mexico), this increase is lower than those reported in other places closer to the Sahara such as Barbados and the Mediterranean. In addition to the impacts on air quality, African dust particles can also be a serious health threat as they serve as the carrier of biological material originating in Africa as reported by Rodriguez-Gomez et al. (2020). If the foreign biological particles are

opportunistic pathogens, they can cause a variety of diseases in the receptor regions, such as the Yucatan Peninsula. Finally, those particles can impact the development of precipitation affecting the regional hydrological cycle when they serve as efficient ice nucleating particles (e.g., Rosinski et al., 1988; Córdoba et al., 2020).

As shown in the present study, combining ground-based off-line and on-line sensors provides robust evidence of the arrival of African dust; however, we also show that the combination of back trajectories with radiosondes, and the estimated surface dust mixing ratio from MERRA-2 are powerful tools that can be exploited when *in situ* information is missing, especially in developing countries where the necessary instrumentation is scarce.

Continuous monitoring of the arrival of African dust is of high importance not only in the Caribbean islands but also at other sites in Latin America such as Mexico, Belize, Guatemala, Honduras, etc. Additionally, epidemiological and statistical studies to track down the number of hospital admissions caused by respiratory issues before and after the arrival of African dust is urgently needed in the Yucatan Peninsula. This will allow policy-makers and local authorities to understand how strong the African dust impact is on local health and the need for better forecasting of such events."

**Reviewer #2:**

General: This paper quantifies the intrusion of African dust in the Yucatan peninsula for the first time and quantifies the impact of this dust intrusion on air quality. This paper is very well-written and should be considered for publication after consideration of my minor comments. Specific Comments

Introduction:

1. Line 45-49: also cite [J. M.; Prospero et al., 2014]

A/ The recommended study is cited in the revised version.

2. Line 55-56: also cite [Barkley et al., 2019; J. M.; Prospero et al., 1981]

A/ The recommended studies are cited in the revised version.

3. Line 76: please replace Prospero et al 2005 with [J. M. Prospero et al., 2002]

A/ The recommended study was replaced in the revised version.

4. Paragraph starting on Line 96, a recent study by Prospero et al 2020 also shows the value of MERRA-2 for tracking African dust transport.

A/ The recommended study is cited in the revised version.

Methods

1. Line 158: Define GoM
A/ Defined (Line 160).

 2. Line 229: HYSPLIT trajectories below 200 m are usually not advised due to the effects of terrain. The authors should provide justification for the 50 m HYSPLIT analysis.

A/ We agree with the reviewer. The 50 m HYSPLIT trajectories were removed. Below the revised Figure is shown.

[Figure]

**Figure S6**. HYSPLIT back trajectories run for 13 days at 250 m (blue) and 500 m (red) agl. Top and bottom trajectories are related to the African dust peaks identified for 2017 and 2018, respectively.

 Results and Discussion

1. Line 304: define pPAHs and NOx.

A/ Defined (Lines 307-308).